# Service Value and Repurchase Intention in the Egyptian Fast-Food Restaurants: Toward a New Measurement Model

**DOI:** 10.3390/ijerph192315779

**Published:** 2022-11-27

**Authors:** Abdelhalim R. Doeim, Thowayeb H. Hassan, Mohamed Y. Helal, Mahmoud I. Saleh, Amany E. Salem, Mohamed A. S. Elsayed

**Affiliations:** 1Social Studies Department, College of Arts, King Faisal University, Al Ahsa 400, Saudi Arabia; 2Tourism Studies Department, Faculty of Tourism and Hotel Management, Helwan University, Cairo 12612, Egypt; 3Hotel Management Department, Faculty of Tourism and Hotel Management, Helwan University, Cairo 12612, Egypt; 4General Management Department, Institute of Management, Economics, and Finance, Kazan Federal University, 420008 Kazan, Russia; 5Graduate School of Management, Saint Petersburg State University, 199004 Saint Petersburg, Russia

**Keywords:** service value, repurchase intention, service equity, service quality, emotional response, reputation, relational benefits

## Abstract

Service value is a crucial dominant indicator in customer decision-making. However, there is a lack of hospitality literature that investigates the multi-dimensional service value in emerging markets. Thus, this study aims to create a multi-dimensional scale for service value and to analyze how different service value dimensions affect customers repurchase intentions at fast-food restaurants. We make a conceptual framework with eight constructs, including service value and repurchase intention. A self-administrated questionnaire is used to gather empirical data from fast-food restaurant customers in Egypt. We employ confirmatory factor analysis to extract the model’s reliability and validity. Moreover, we use a structural equation model to extract the model regressions and correlations using AMOS software. We find that each of the eight proposed service value variables impacts fast-food restaurant customers’ repurchase intention. However, the factors that strongly influence customers’ preferences to make more purchases are service equity, confidence benefits, service quality, and service reputation. We contribute to the literature on hospitality customer value and repurchasing intentions by presenting a comprehensive multi-dimensional service value framework that affects customers’ repurchase intentions in fast-food restaurants. Practically, eight service value variables can help managers of fast-food restaurants meet customer needs and gain a competitive advantage. We suggest many crucial recommendations to restaurant managers regarding the priority of the service value constructs. For example, managers should consider service equity, service quality, and service reputations as a priority of the restaurant service value.

## 1. Introduction

With the exponential growth of restaurant types and franchised restaurants, restaurants have embraced new strategies to enhance service value and maintain customers’ repurchase intentions [1]. Repurchase intention has led to restaurant brands’ growth, as it refers to the customer’s tendency to re-try/re-buy the same services several times with the same service providers [2]. Repurchase intention raises service providers’ revenues and improves their reputation. Repurchase intentions occur when most restaurants have begun enhancing service value machinery to manage customers’ services [1]. Liang et al. [3] and Hsu et al. [4] indicated the determinants of repurchasing intention; they found that the repurchasing intention toward services results from customers’ satisfaction and trust toward service providers.

Additionally, some factors established repurchase intentions, such as service providers’ plan quality, system quality, information quality, knowledge content quality, perceived usefulness of service knowledge sharing, and eliminating customer expenditures (e.g., prices and costs) [5]. With customers’ growing expectations for positive service experiences, restaurants cannot rely merely on the previous factors that led to repurchase intention. Still, they need to enhance the service value of customers proficiently and better understand their needs. 

The restaurant’s service value is the fundamental purpose of managers to help restaurants achieve profits, customer satisfaction, and loyalty [6]. Thus, customers’ value explains their behavioral intentions; the value refers to the evaluation of the offered utilities regarding what is given and received [7,8]. In this sense, customers’ value of offered services or service value as an entity is a holistic construct that affects customer experiences in restaurants. Moreover, service value provides a conceptual lens through which customers can evaluate the restaurant experience [6]. Service value is the interaction between customers and service providers, leading to evaluative and multisensory consequences from contextual factors (e.g., time, place, employees, atmosphere) [9]. 

Furthermore, restaurant service value reflects the accumulation of the seeking, expecting, or experiencing benefits [7]. However, there is a lack of literature investigating the determinants of service values, still vague in the restaurants’ context, especially on repurchase intentions. Moreover, some other dimensions relate to service value, such as Service relational benefits: relational service benefits are a part of service relational benefits that benefit customers as separated from the core services. The definition means regular customers will likely appreciate extra benefits because of their long-term adherence to service providers and special and social benefits [10]. Service equity: service equity refers to providing service to all customers without discrimination and disparities in the delivery to promote safety and service quality [11]. As for the dimensions of service value, Petrick [12] provided dimensions that affect the service values and repurchase intentions: Emotional response: emotional responses refer to physiological, cognitive, intense, brief, and mental reactions to individuals’ behavior regarding various situations [13]. Behavioral price and monetary price: behavioral responses to the prices refer to the interaction of customers toward the prices of services considering customers’ experiences, needs, and service value [14]. Service quality: service quality refers to the differences between customers’ expectations and perceptions of specific services regarding their experiences and the full-service values [15]. Service reputation: service reputation is a public/private perception of the service providers and how it operates their services [16]. 

For the privilege of service value dimensions and their impacts on customer behavioral intentions in restaurants, the current study sheds light on two gaps in the hospitality management research related to customer value in restaurants, and we contribute to the current literature as follows: (a) Even though factors that drive customers to repurchase intentions are widely investigated in hospitality [7,12], there is a lack of investigations of all antecedents of perceived service value in the service choice repurchase intentions in fast-food restaurants. Thus, this study aims to fill the current gap by providing new dimensions essential to investigating antecedents of customer repurchase intentions in restaurants (i.e., behavioral price, emotional response, service quality, service reputation, service equity, confidence benefits, social benefits, and special treatment benefits). (b) Research indicates that customers’ perception of services involves cognitive components of their emotions that affect their intentions. Thus, we emphasize extending the theoretical framework by studying the customers’ emotions and behaviors during service encounters at restaurants, especially price behavior and emotional responses, with different influences on their repurchase intention. We organize our paper as follows: the first section is the introduction, followed by the literature review. After that, we conduct the methods and material part; afterward, we examine the results; finally, we illustrate our discussion, implications, and further research.

## 2. Literature Review

### 2.1. Service Value 

Customers’ acceptance of restaurant services differs depending on the perceived service value [10,11]. Accordingly, restaurants continuously learn to deliver the best service value to their customers [17]. Service value refers to the customer’s complete judgment of the service’s usefulness based on perceptions of what is received and given [18]. In line with the previous definition, Babin and James [19] defined service value as a method of evaluating a service after it has been utilized. Hence, the perceived service value highlights that a service’s value can only be judged subjectively from the customer’s perspective [20]. In this study, perceived service value refers to the value that customers experience after being helped through the full service or self-service process and experiencing a higher level of satisfaction than before.

Several studies showed service value as a one-dimensional structure [21,22,23]. The one-dimensional structure of service value is simple, but it misses the construct’s conceptual richness [24]. However, Ruiz et al. [24] emphasized using a constructive approach to service value models. Although they discovered substantial support for such a formative conception of value, their study is limited because the list of service value components they examined may not be complete. For example, in addition to confidence, relational benefits such as social and special treatment may be meaningful in contexts characterized by intense, personal relationships between providers and customers [25]. Following the approaches of Zeithaml [18] and Petrick [12], we want to comprehend the service value by studying the customers’ perspective on the benefits they have provided or obtain the weight in exchange for their sacrifices regarding their behavioral intentions.

### 2.2. Behavioral Intentions

Behavioral intentions consider one of the strongest tendencies for making decisions [26]. The intention is the attitude that stimulates individuals to act or react to specific events or services [27]. The tourism and hospitality literature has evidence that intentions are double-promoted inside tourists/customers, meaning that intention could be antecedents to further decisions and consequences of different stimulations [26]. For instance, intentions could be antecedents to re(visit) destinations or repurchase services. It could also be a consequence of the positive, memorable experiences during service encounters or service providers’ infrastructure, superstructures [28], and customers’ reviews [29]. Importantly, intentions do not guarantee the actions; in other words, the customer may have the intention to do something, but the action may be delayed or postponed. According to Wang and Li [27], some factors affect the real action beyond customers’ intentions, such as perceived environmental knowledge, pro-environmental purchase intention, social pressure, poor marketing, lack of distribution, and time pressure. Thus, behavioral actions and intentions are not equivalent. 

Repurchase intentions are among the most crucial antecedents of tangible actions made by customers, leading to service providers’ revenue growth [1]. Hospitality and tourism scholars highlighted some factors that drive tourists to repurchase their intentions, such as customer satisfaction [1]; customer trust, commitment, and perceived value [30]; customer equity by enhancing the service values [5]; all related consumption values (e.g., emotional value, social values, and situational values of services) [31]. 

One significant claim that affects customer behavioral intentions (repurchase intention) with subsequent decisions to take action is the customer’s mood toward service encounters [26]. Customer mood is difficult to predict because its internal states emerge without clear causes [27]. However, some avital predictors may lead to understanding customers’ moods and behavioral intentions that directly relate to customers during service encounters [26,32]. The following sections illustrate the most prominent factors that directly affect customer behavioral intentions, mainly restaurant repurchase intentions.

#### 2.2.1. Behavioral Price and Repurchase Intentions

Pricing considers one of the most crucial signaling instruments and managerial decision-making. Thus, customers’ behavioral responses to the pricing policies imposed by service providers are essential to predicting customer intention toward (re)using/purchasing the services. A recent study by Kalyanaram and Winer [14] found a strong relationship between the prices as a stimulus and the intentions as behavioral outcomes regarding these stimuli. For instance, when customers tend to behave differently toward services in terms of price fairness, they seek to avoid the services where the price has substantial variations than stable prices [33]. Additionally, some service providers prefer to build their price strategy on behavior-based pricing (BBP), which refers to how service providers gather customers’ purchase history data, analyze their behavior, and then offer variances [34]. However, there is a cautionary warning for the BBP, as it may cause unfair prices, leading customers to be dissatisfied with service providers [35].

Moreover, customers have different behaviors regarding the reference price point, meaning that they attribute the prices to internal or external norms representing the aggregate effects of their experiences (present and past stimulations) [36]. According to Kalyanaram and Winer [14], customers who refer to internal norms as a reference point are more sensitive and have different behavioral outcomes than external reference norms of prices. Understanding how customers behave toward prices is essential because different behavioral outcomes regarding customers’ needs affect their intentions to use or purchase the services [37]. Furthermore, as mentioned earlier, different prices led to different behavioral outcomes, and behavioral outcomes stimulated the intentions. Therefore, we can hypothesize that: 

**H1.** *Positive pricing behavior perception significantly influences customers’ repurchase intentions at restaurants*.

#### 2.2.2. Emotional Response and Repurchase Intentions

Emotions refer to intense, brief, mental, and physiological reactions regarding different situations. Emotion definitions include elements of measurement levels (behavioral, cognitive, and physiological) and emotion processing (explicit and implicit) [13]. Both emotion measurement levels and emotion processing refer to different approaches according to Kaneko and colleagues’ [38] assumptions: (a) implicit emotional responses: responses triggered automatically or indirectly by the stimulus itself without conscious awareness; (b) explicit emotional responses: responses that happen with conscious effort and have the level of awareness; (c) behavioral, emotional responses: responses that refer to the implicit and/or explicit responses relationship to the customers’ bodies (e.g., amount consumed and face and body movement); (d) physiological, emotional responses: largely implicit responses that refer to activities of the bodily functions and autonomic nervous system; (e) cognitive, emotional responses: explicit decisions and choices that may unknowingly be affected by implicit emotional responses [39].

Customers make decisions in the food and beverage industry based on the conceptualizations of services offered rather than just the food, because emotions are more closely linked to the conceptualizations of services [13]. For instance, when customers experience an unbranded piece of dark chocolate in a restaurant, their emotional responses do not react to the dark chocolate itself. However, their emotional responses come from associating it with dark chocolate’s conceptualizations (emotional association: relaxing and calm; functionality: sugary and fattening) more than its sensory characteristics [40]. Regarding the importance of the emotional response to customer behaviors, Low et al. [41] illustrated that customers’ behavioral responses differ from their emotional responses to the service provided. They found that enthusiasm, emotional engagement, and interested emotion had slightly higher rates in cafés than in booths, whereby café customers encounter many services and interact with them. Given the previous arguments about how positive emotional responses drive customer decision-making and customer intention is a post hoc decision-making process, we can hypothesize that: 

**H2.** *Positive emotional responses positively influence repurchase intentions at the restaurant*.

#### 2.2.3. Service Quality and Repurchase Intentions

Service quality in restaurants is an essential element influencing customer satisfaction and behavioral outcomes. Service scholars have defined service quality as the variance between customers’ perceptions and expectations during service encounters [15,42,43]. In other words, we can measure service quality by comparing their behavioral-based expectations and their perceptions of the encountered services. Thus, positive service quality refers to the received services being similar to or exceeding the customers’ expectations. On the contrary, the negative gap in service quality relates to the customer perception of not meeting their expectations [15,44]. 

Service quality could be an antecedent or consequence as well. On the one hand, regarding service quality as an antecedent, we may consider service quality as the antecedent of customer behavioral outcomes. According to Namin et al. [45], service quality is the navigator for customers’ behavioral outcomes during service encounters. Customers who perceive negative service quality will likely become confused and prefer not to reuse or repurchase the service [46]. In contrast, when customers find that service providers care to introduce high-quality services, they are likely to have positive behavioral outcomes toward service providers [45]. 

On the other hand, we may consider service quality perception due to customers’ previous experiences in restaurants [46]. Thus, experienced tourists may have different perceptions of service quality than customers who do not have experiences. We may also consider it because of service providers’ initiatives to enhance customer experiences. For instance, Ha and Jang [44] highlighted different causes for positive service quality (e.g., menu design, food variety, restaurant atmosphere, serving size, product quality, food taste, freshness, healthy options, and temperature to assess food quality). Furthermore, some crucial factors lead to restaurants’ positive service quality, such as (a) restaurant interior (e.g., place ware and eating utensils); (b) restroom (e.g., cleanliness of the floor, slippers, and availability of toilet paper); (c) employees’ efficiency and hygiene (e.g., waitstaff hair and hands, and waitstaff uniforms) [47]; building exterior (e.g., trash and cigarettes) [48]. High service quality drives customer satisfaction, and satisfaction leads to establishing repurchase intentions. Therefore, we can hypothesize that: 

**H3.** *High service quality positively influences customer repurchase intentions at restaurants*.

#### 2.2.4. Service Reputation and Repurchase Intentions

Reputation describes the significant service attractiveness of firms compared to other leading competitors [49]. According to Roig et al. [50], a firm’s service reputation is vital in producing a desirable value for its customers. Customers consider a restaurant’s service reputation when choosing a restaurant; hence, it should provide them with a unique experience [51]. Well-known restaurant brands improve their service value reputation by incorporating features to produce enticing visual stimuli, brand awareness, emotional linkages with pleasure, and sociability [52]. Roper and Fill [53] defined service reputation as an indicator of a firm’s value. The current study defines service reputation value as the services that restaurant customers talk about, distinguishing the restaurant and creating customer value.

Increasing trust and generating customer value are based on service reputation [12,54]. A previous study showed that reputation directly impacts customer satisfaction and loyalty [54]. Qalati et al. [54] indicated that a positive reputation communicates a low-risk level and encourages purchasing decisions. One of the essential factors in accommodating visitors’ decision-making is reputation [55]. According to Boo [56], suggestions from friends, the restaurant’s current reputation, and perceived value may all significantly impact customers’ restaurant choices. Oh [57] mentioned that one of the most important antecedents of repurchase intentions is reputation. Therefore, we can hypothesize that: 

**H4.** *Service reputation value positively influences repurchase intention*.

#### 2.2.5. Service Equity and Repurchase Intentions

Another aspect of service value is service equity. Service equity is also known as service image or brand equity [24]. Service equity is defined as the effect of brand knowledge on how customers react to a brand’s marketing [58]. A brand’s service equity is likely to be positive if it receives more positive reactions [59]. The service equity concept describes a relationship between a restaurant and its customers to improve the restaurant’s competitiveness and build long-term value in the eyes of the customers. Accordingly, customers’ decisions about restaurant choices are influenced by substantial restaurant service brand equity, ensuring service quality, food quality, pricing, and standards [60]. 

Furthermore, Ruiz et al. [24] stated that service equity is a source of value creation as restaurant communications and customers’ interactions with the service impact their impressions of the restaurant. A strong brand generates feelings of intimacy, affection, and trust and significantly affects customer value perceptions [61]. Hashim and Haque [62] discovered that the service equity of resort customers significantly impacted their repurchase intention to stay at resorts. Therefore, the current study contends that service equity provides value to customers by offering a consistent and powerful value proposition and customer experience that will satisfy customers and keep them coming back. Therefore, we can hypothesize that: 

**H5.** *Service equity value positively influences repurchase intention*.

#### 2.2.6. Relational Benefits and Repurchase Intentions

Customers seek desired benefits that satisfy their needs when using restaurant services [25]. These benefits contribute to developing and maintaining long-term customer relationships, reducing the cost of gaining new customers, and providing customer value [63]. The perceived value of the restaurant’s relational benefits significantly impacts the strength of a customer’s relationship with the restaurant [64]. Relational benefits are “the long-term value that customers obtain from the firm in exchange for their relationship” [65]. This definition implies that frequent restaurant customers will likely receive additional benefits because of their long-term commitment to a restaurant [66,67]. This study defines rational benefits as the value that the customer derives from the benefits of dealing with the restaurant.

There are three dimensions to relational benefits (i.e., confidence, social, and special treatment benefits) [64]. First, confidence benefits refer to the comfort or feeling of security the customer receives from the restaurant’s services, so the customer earns a high value [24]. Confidence benefits allow customers to have excellent long-term services, a low-risk perception, increased confidence in the restaurant’s reliability and integrity, and to make correct predictions about future service experiences [68]. Secondly, social benefits reflect the intensity of forming personal bonds between customers and the restaurant, customers’ experience with the restaurant, and the restaurant’s recognition of customers [31]. Hence, social benefits are vital in establishing customer value by forming social bonds through long-term relationships between restaurants and customers [67]. Finally, special treatment benefits indicate the personalized service geared to fulfil customers’ specific needs (e.g., extra specialized service), more focus, and special services not commonly provided to other customers. The underlying motivator for customers to form and sustain connections with service providers is the promise of special treatment [66]. Therefore, this preferential service gives customers the value they require while also making them feel unique, valued, and appreciated [64,69].

According to previous research, customers expect benefits from service providers, such as confidence, which lowers perceived risk and anxiety and boosts customer value [70,71]. As customers participate in relationship behavior and accrue experiences with the service, their level of uncertainty diminishes as their understanding of the service provider grows [24]. Moreover, social benefits encourage customers to form emotional relationships with restaurants [67]. Hence, emotionally attached customers establish an affective attachment to a restaurant and exercise positive reciprocity by offering referrals and avoiding moving to another [72]. In addition, customers believe special treatment from service providers is necessary for building long-term partnerships and creating customer value [73]. As a result, positive repurchase intentions, such as word of mouth and return visits, are linked to customer perceptions of these relational benefits [74]. Therefore, we can hypothesize that: 

**H6.** *Confidence benefits positively influence repurchase intention*.

**H7.** *Social benefits positively influence repurchase intention*.

**H8.** *Special treatment benefits positively influence repurchase intention*.

Collectively, an illustration of the main hypotheses of the current study is shown in Figure 1.

## 3. Materials and Methods

### 3.1. Constructs Measures

The study variables were taken from prior studies to guarantee the constructs’ content validity. We developed our variable by adopting the following constructs from Petrick [12]: behavioral price (4 items), emotional response (5 items), service quality (4 items), and service reputation (5 items). Moreover, service equity (4 items), confidence benefits (5 items), social benefits (4 items), special treatment benefits (5 items), and repurchase intentions (3 items) were the other constructs adopted from Lee et al. [64] and Ruiz et al. [25]. These variables were developed by considering our level of analysis, “restaurant level”, and word consistency remained the main latent variable understandable for customers.

### 3.2. The Study Context and Data Collection

We distributed our surveys in Egypt. However, Egypt has no publicly accessible database or report on fast-food restaurants [75]. We employed convenience sampling to choose respondents from fast-food restaurant customers. The data were gathered from fast-food restaurant customers in Cairo, Egypt. We distributed the survey based on frequent customers who visited fast-food restaurants in Cairo; customers were freely asked to participate in the study. There were two main sections to the questionnaire. A fast-food restaurant customer profile is presented in the first section. Section two contains a 5-point Likert scale used to evaluate all the study’s 25 items (1 = “strongly disagree” to 5 = “strongly agree”). The questionnaires were distributed between May 2022 and August 2022. Four hundred questionnaires were distributed, and three hundred and forty-three (*n* = 343) valid questionnaires were completed and returned, resulting in an 85.7% response rate. 

### 3.3. Data Analysis

Exploratory factor analysis was utilized to evaluate the internal consistency of the measures and establish their factor structure. Following exploratory factor analysis, confirmatory factor analysis was utilized to determine whether the manifest variables in nine constructs with multiple-item scales represented the hypothesized latent variables. The construct reliability was tested using composite reliability (CR) and Cronbach’s alpha for each latent variable, and the constructs convergent validity and discriminant validity were tested using the average variance extracted (AVE) (Hair et al., 1998 [76]). Standardized path coefficients (ß) were used to test the proposed hypotheses after the measures were confirmed.

## 4. Results

### 4.1. Sample Profile

According to Table 1, 46.9% of men and 53.1% of women were among the respondents. In addition, 72% of participants were between the ages of 18 and 39, while just 28% of respondents were 40 or older. There were 46.6% of respondents with a bachelor’s degree, 34.7% had only completed their secondary education, and 18.7% had only completed their postgraduate studies. Regarding marital status, the respondents’ percentages were 50.1% single, 27.4% married, and 22.4% married with children.

### 4.2. Measurement Model

Table 2 depicts the reliability test (i.e., Cronbach’s alpha) for each construct tested between 0.93 and 0.98, above 0.70, indicating the reliability of all variables [77]. All the constructs have substantial internal reliability, as seen by the constructs’ composite reliability ranges from 0.95 to 0.98 [76]. The structures comprised every item with factor loadings greater than 0.50 [77]. Regarding discriminant validity, each construct’s AVE is higher than the squared correlations between the components (see Table 3) [78].

The fit of the measurement model was evaluated using various fit indices. The total model chi-square with 81 degrees of freedom was 150.160 (*p* < 0.001). The adjusted goodness-of-fit (AGFI) value was 0.93, the goodness-of-fit (GFI) value was 0.92, the comparative fit index (CFI) value was 0.96, and the relative/normed chi-square (2/df) value was 1.853. Findings from fit indices indicated a satisfactory model fit [78].

### 4.3. Hypothesis Testing

As shown in Table 4 and Figure 2, all hypotheses were supported by the investigation of the path coefficients. Each of the predictions positively impacted the repurchase intentions of fast-food restaurant customers. However, it is notable that service equity (β = 0.803, *p* ≤ 0.000), confidence benefits (β = 0.532, *p* ≤ 0.000), service quality (β = 0.455, *p* ≤ 0.000), and service reputation (β = 0.400, *p* ≤ 0.000) all had significant effects on customers’ intentions to make more purchases. After that, the other service value factors, including emotional responses (β = 0.323, *p* ≤ 0.000), behavioral price (β = 0.275, *p* ≤ 0.000), social benefits (β = 0.274, *p* ≤ 0.000), and special treatment advantages (β = 0.234, *p* ≤ 0.000), all had nearly similar effects on customers’ intentions to make another purchase.

## 5. Discussion and Implications

### 5.1. Discussion

Once restaurants apply service value strategies, repurchase intentions will develop [1]. Restaurants cannot only rely on the primary variables (i.e., behavioral and monetary price, emotional response, quality, reputation, equity, and confidence benefits) [12,24] that lead to repurchase intention, because customers’ expectations for great service experiences are growing. However, they still need to improve customer service value and better comprehend their needs. Therefore, this study attempts to fill the current gap by adding additional aspects necessary for analyzing the causes of customers’ intent to make repeat purchases at fast-food restaurants (i.e., social benefits and special treatment benefits).

One of the crucial factors in a customer’s choice to repurchase is the price [34]. This study showed that customers’ perceptions of restaurants’ favorable pricing behavior significantly impact their intentions to make repeat purchases. This result is consistent with a recent study by Kalyanaram and Winer [14], which discovered a substantial correlation between the intentions as behavioral outcomes for these triggers and the pricing as a trigger. Therefore, fast-food establishments must comprehend how customers interact with prices because various behavioral outcomes relating to customer demand impact customers’ intentions to utilize or purchase services [37].

Emotional responses are the ability to recognize affective stimuli by displaying emotion [38]. Positive emotional responses increased repurchase intentions at the restaurant in this study. This finding supports Damasio’s [79] assertion that emotions guide decision-making. Additionally, our findings are consistent with those of Petrick [12], who discovered that emotional response is a crucial component of service value and influences customers’ intentions to make additional purchases. Fast-food restaurants ought to be concerned about the emotions of their customers and should use brand messages to help them do so [38].

Customer behavior during service interactions is steered by service quality [45]. According to this study, excellent service quality at restaurants influences customers’ intentions to make additional purchases. As a result, our findings supported the hypothesis by Namin et al. [45] that customers are more likely to behave favorably toward service providers when they perceive them to be concerned about introducing high-quality services. Fast-food establishments should, therefore, be concerned with factors that contribute to service quality, such as speed of service, food variety, restaurant ambience, serving size, product quality, food taste, and freshness [44]. Therefore, one of the crucial elements of service value and a reliable catalyst for customer repurchase is service quality.

A restaurant’s service reputation is crucial in creating a desirable value for its customers [50]. This study discovered a favorable relationship between service reputation value and repurchase intention. This finding is in line with Oh [57] and Qalati et al. [54], who both stated that reputation is one of the most significant predictors of repurchase intentions. Additionally, our findings are consistent with Petrick’s [12] assertion that building customer value is built on service reputation. Therefore, fast-food restaurants should constantly improve their reputation for providing high-quality service by adding elements that create appealing visual stimuli, brand recognition, emotional links to pleasure, and sociability [52].

Substantial restaurant service equity impacts customers’ choice of restaurants [60]. According to this study, service equity value positively influences repurchase intention. This outcome is consistent with what Hashim and Haque [62] found: that resort customers’ service equity had a significant impact on their intention to repurchase resort stays. Additionally, this outcome supports the assertion made by Ruiz et al. [24] that service equity is a source of value creation as customer interactions and restaurant communications affect how customers perceive the establishment. Fast-food restaurants should, therefore, consistently improve service equity by providing inclusive and equitable service to all customers [60].

Providing rational benefits helps create and maintain lengthy customer relationships, lowers the expense of acquiring new customers, and adds value to customers [63]. This study’s findings supported the notion that social benefits, confidence, and special treatment benefits all positively influence repurchase intentions in fast-food establishments. This finding is congruent with the findings of Gwinner et al. [74], Ryu and Lee [64], and Gupta [68], who discovered that positive repurchase intention, such as word of mouth and return visits, is related to customer perceptions of these relational benefits. Our findings also support other studies claiming that customers anticipate service providers’ benefits. For instance, social benefits encourage customers to form emotional bonds with restaurants [67]; confidence benefits lessen perceived risk and anxiety [70]; special treatment benefits from service providers forge long-term partnerships and increase customer value [73]. Therefore, fast-food establishments should continuously enhance the rational benefits of their customers. For example, lower risk and anxiety, foster more customer trust, enhance social ties with customers, provide quick service to all customers, especially regulars, offer discounts and specials, and learn about loyal customers’ preferences [63,67,73].

### 5.2. Theoretical Contribution 

This study contributes to customer experience in the hospitality industry in several ways. First, we outpace the traditional restaurant service value approach by modelling service value dimensions as a set rather than limited conceptions influencing customer repurchase intention. The current study contributes to the restaurants’ service value literature [10,12,24,25] and repurchasing intentions in restaurants [7,12,31] by providing the service value dimensions (i.e., behavioral price, emotional response, service quality, service reputation, service equity, confidence benefits, social benefits, and special treatment benefits) influencing on repurchase intentions in fast-food restaurants.

Second, we add to the literature service value dimensions of both constructs, namely social benefits and special treatment benefits, to contribute to Ruiz et al.’s (2008) [24], which focused only on confidence benefits as a sub-category of relational benefits in fast-food restaurants. By including these two factors in the service value measure, restaurants can learn more about the extent and nature of the social relationships between customers and how they feel about the level of special services they receive [32,78]. Thirdly, we found that customers in fast-food restaurants care more about service equity, confidence benefits, service quality, and service reputation than about behavioral price, emotional response, social benefits, and special treatment benefits. Thus, we investigated the relations and studied how the strengths of different dimensions affect customer repurchase intentions.

Finally, in the literature on service value in restaurants, experience measures the product attributes (e.g., the quality of the service and food, in addition to the atmospherics of the restaurants) [7]. However, our approach in the current study measures how customers behave differently toward the restaurant, considering their emotional responses, service reputation, and behavioral price. Thus, we shift attention from studying the value from the perspective of the product to the customer value perspective, providing insightful managerial implications.

### 5.3. Managerial Implications

This study assists managers of fast-food restaurants by offering a comprehensive model for evaluating the service value and how this model affects customers’ intentions to make repurchases. Managers should consider all the service value model components when determining the degree of service value at the fast-food restaurant. Fast-food restaurants managers should understand how customers interact with products and services prices [37]; the emotions of their customers [38]; required service quality features (e.g., speed, variety, ambience, time, taste, and freshness) [44]; level of restaurant reputation [52]; providing inclusive and equitable services to the customer [60]. Additionally, fast-food restaurant managers should continually improve their customers’ rational benefits (e.g., reduce risk and anxiety, strengthen relationships with customers, and offer discounts and promotions) [63,67,73].

Another suggestion is that fast-food restaurant resource allocation may vary depending on customers’ perceptions of service value. According to the study’s findings, fast-food restaurant customers are more concerned with service equity, confidence benefits, service quality, and service reputation than other factors such as behavioral price, emotional response, social benefits, and special treatment benefits. Therefore, managers of fast-food restaurants should assign specific messages to customers regarding service equity, confidence benefits, service quality, and the service reputation of a fast-food restaurant in their advertising campaigns and technology platforms. Furthermore, customers’ service value requirements may differ from one restaurant to the next. As a result, managers of fast-food restaurants can employ the model provided in this study to influence customers’ return to the restaurant.

### 5.4. Limitations and Further Research

This study has certain limitations, which opens up possibilities for further research. This research aims to develop a comprehensive service value model for the hospitality industry. The model was tested empirically in fast-food restaurants. A future study could test the proposed model in different hospitality establishments (e.g., hotels, resorts, cafes, and other restaurant types). This study collects data from restaurant customers using a self-administered questionnaire. As a result, more studies might investigate our model using qualitative approaches (e.g., focus groups and interviews) or a combination of mixed methods (i.e., quantitative and qualitative). The customer value is evaluated and viewed from the customer’s perspective. As a result, more research on the impact of demographic variables (e.g., age, gender, income, generations, and educational attainment) on the service value model would be beneficial.

Moreover, this study’s participants were customers of fast-food restaurants in Greater Cairo, Egypt. As a result, future studies could include cross-country comparisons in the service value model to improve the generalizability of the findings. As previously indicated, this study gathered data from customers’ viewpoints, and it is critical to obtain restaurant management’s thoughts on the proposed service value framework.

## 6. Conclusions

In the current study, we aimed to carry out a multi-dimensional scale for service value to assess the impact of different service value attributes on consumers repurchase intentions at fast food restaurants. Based on a structural equation modelling technique, we showed that eight service value domains were significant predictors of customers repurchase intentions, with the highest impact exerted by service equity, confidence benefits, service quality, and service reputation. Therefore, we add to the existing literature regarding the effect of service value framework in the hospitality industry, in which the mentioned empirical findings could help the managers of fast-food restaurants gain a competitive advantage.

## Figures and Tables

**Figure 1 ijerph-19-15779-f001:**
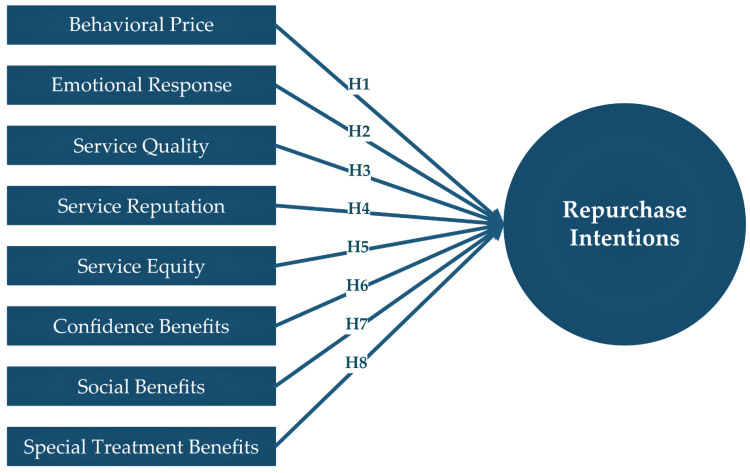
The conceptual framework.

**Figure 2 ijerph-19-15779-f002:**
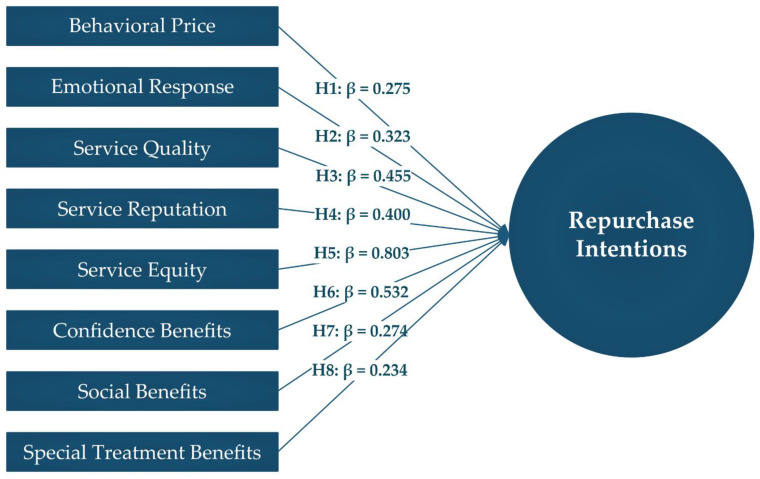
The structural model.

**Table 1 ijerph-19-15779-t001:** Sample profile.

Characteristics	Descriptions	Statistics	(%)
Gender	Male	161	(46.9)
	Female	182	(53.1)
Age	18–28	130	(37.9)
	29–39	117	(34.1)
	40 or more	96	(28.0)
Education	Secondary school or below	119	(34.7)
	University degree	160	(46.6)
	Postgraduate (Diploma–Master–PhD.)	64	(18.7)
Marital status	Single	172	(50.1)
	Married	94	(27.4)
	Married with children	77	(22.4)

**Table 2 ijerph-19-15779-t002:** Measuring model analysis.

Constructs	AVE	Composite Reliability	Cronbach’s Alpha
Behavioral price	0.92	0.94	0.97
This restaurant service required little energy to purchase			
This restaurant service is easy to shop for			
This restaurant service required little effort to buy			
This restaurant service is easily bought			
Emotional response	0.93	0.93	0.98
This restaurant service makes me feel good			
This restaurant service gives me pleasure			
This restaurant service gives me a sense of joy			
This restaurant service makes me feel delighted			
This restaurant service gives me happiness			
Service quality	0.93	0.91	0.98
This restaurant service is outstanding quality			
This restaurant service is very reliable			
This restaurant service is very dependable			
This restaurant service is very consistent			
Service reputation	0.92	0.96	0.96
This restaurant service has a good reputation			
This restaurant service is well respected			
This restaurant service is well thought of			
This restaurant service has the status			
This restaurant service is reputable			
Service equity	0.83	0.87	0.95
It makes sense to buy this restaurant’s services compared to others, even if they are the same.			
Even if another restaurant offers the same service, I would still prefer this restaurant.			
If another restaurant offers services as good as this company’s, I would still prefer this restaurant.			
If another restaurant is not different from this restaurant, it still seems more intelligent to purchase this restaurant’s services.			
Confidence benefits	0.94	0.89	0.98
I have more confidence that the service will be performed correctly.			
I have less anxiety when I buy/use the services of this restaurant.			
I believe there is less risk that something will go wrong.			
I know what to expect when I go to this restaurant.			
I feel I can trust this restaurant.			
Social benefits	0.96	0.90	0.93
This restaurant staff recognizes me			
This restaurant staff know my name and official title			
This restaurant staff treat me as his family			
I enjoy certain social aspects of the relationships			
Special treatment benefits	0.91	0.93	0.94
Even though I did not talk to this restaurant management. I get free food and beverage service			
I get faster service than most customers			
I get discounts or special deals that most customers do not get.			
I get the preferred seat I usually use when visiting this restaurant.			
They do service for me according to my food and beverage taste.			
Repurchase intentions	0.89	0.84	0.96
I intend to continue doing business with this restaurant in the future.			
As long as the present service continues, I doubt I will switch restaurants.			
I will choose this restaurant the next time I need this service.			

**Table 3 ijerph-19-15779-t003:** Discriminant validity.

Constructs	1	2	3	4	5	6	7	8	9
Behavioral price	0.95								
Emotional response	0.93	0.96							
Service quality	0.92	0.95	0.96						
Service reputation	0.89	0.93	0.94	0.95					
Service equity	0.88	0.91	0.89	0.82	0.91				
Confidence benefits	0.82	0.95	0.88	0.92	0.85	0.94			
Social benefits	0.92	0.94	0.95	0.93	0.81	0.93	0.97		
Special treatment benefits	0.91	0.95	0.93	0.91	0.90	0.91	0.95	0.95	
Repurchase intentions	0.82	0.86	0.84	0.75	0.77	0.79	0.74	0.90	0.94

Note: All correlations are significant at *p* < 0.001. Note: 1: Behavioral price; 2: Emotional response; 3: Service quality; 4: Service reputation; 5: Service equity; 6: Confidence benefits; 7: Social benefits; 8: Special treatment benefits; 9: Repurchase intentions.

**Table 4 ijerph-19-15779-t004:** Path coefficients.

Hypotheses	Path	Beta (ß)	*t*-Values	*p*-Value
H1	Behavioral price → Repurchase intentions	0.27	7.51	<0.0001 ***
H2	Emotional response → Repurchase intentions	0.32	3.64	<0.0001 ***
H3	Service quality → Repurchase intentions	0.45	7.72	<0.0001 ***
H4	Service reputation → Repurchase intentions	0.40	5.36	<0.0001 ***
H5	Service equity → Repurchase intentions	0.80	26.36	<0.0001 ***
H6	Confidence benefits → Repurchase intentions	0.53	16.44	<0.0001 ***
H7	Social benefits → Repurchase intentions	0.27	3.59	<0.0001 ***
H8	Special treatment benefits → Repurchase intentions	0.23	3.82	<0.0001 ***

*** statistically significant at *p* < 0.0001.

## Data Availability

Data are available on request due to privacy/ethical restrictions.

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
