# Peer review of "Service Value and Repurchase Intention in the Egyptian Fast-Food Restaurants: Toward a New Measurement Model"

_ijerph, 2022, doi:10.3390/ijerph192315779_

Round 1
Reviewer 1 Report
Dear Authors,
Thank you for the opportunity to read your article, " Service value and repurchase intention in the Egyptian fast-food restaurants: towards a new measurement model" I thoroughly enjoyed reading your article, and I believe you are addressing a timely and relevant issue. However, after reading your research, I had some significant reservations about it, which prevented me from recommending its publication. I believe that the editors should reconsider this article after some major revisions. I'll go into more detail about my concerns below. This response letter is structured in the same manner as your article. I hope these suggestions will assist you in improving the quality of your research.
Abstract
The abstract is currently not self-explanatory. I suggest revising the abstract to make it more effective and compelling in reporting the unique value of this research and its contribution to theory and practice. First, the goal is unclear. Second, the section on methods is not effectively reported. Finally, the recommendations are not discussed.
Introduction
I believe the introduction leaves out several essential aspects. The authors do not effectively position their research within the current scholarly debate. The authors must clearly position their paper in the scholarly literature, emphasizing how it will contribute to the scientific debate. Improving the article's positioning should lead the authors to identify the knowledge gaps affecting the scholarly debate. At the moment, it is unclear what scientific debate the authors intend to address with their research. The current version of the introduction provides insufficient information to pinpoint the originality and relevance of this research. Furthermore, the authors do not provide a critique of the existing scales and do not explain why a new scale is even necessary.
Literature Review
Authors primarily focus on hypothesis development and various aspects of the scale to be developed in the literature review. However, they make no mention of the scales already in use to measure the variables of interest, their shortcomings, or why a new measurement model is required. Furthermore, the review of literature lacks a global dimension and omits several important studies. Please include the following literature into the study:
https://doi.org/10.1016/j.jneb.2014.01.002
https://doi.org/10.3390/ijerph18031175
https://doi.org/10.1016/j.landusepol.2020.105250
https://doi.org 10.1111/cjag.12184
https://doi.org/10.1016/j.jada.2004.08.030
https://doi.org/10.1016/j.jretconser.2021.102556
Materials and Methods
The methods section is extremely lacking. Many details are left out, such as how the variables or constructs were developed, how they were contextualized and adapted, and what methods were used to measure relibality and validity. Do the authors employ inter-rater reliability or other techniques? What is the justification for any methods chosen? Furthermore, the details of the data collection process are hazy, and the method is not scientific. I recommend that the authors include as many details as possible to improve the clarity of this section.
Results
The results section is similarly disorganized. The findings are not presented in a way that justifies the study's objectives. Simply including some tables without describing the context or purpose is not helpful to the reader. Please improve the results section and describe in detail how your measurement model evolved and what contribution it makes.
Discussion and Conclusion
The general discussion merely presents an overview of the study findings, but it does not critically contextualize the study results in light of the extant scholarly debate. As a consequence, the authors are unable to provide us with thick and consistent information about how they are adding to the scholarly debate. Implications for theory and practice are limited. The authors do not provide adequate insights about how we can advance the scholarly knowledge about the new measurement model being proposed. Besides, they do not provide us with compelling management implications supporting decision making processes in the food industry.
Finally, yet importantly, the authors do not effectively elaborate on a relevant and inspiring agenda for further developments.
Author Response
Responses to Reviewer two
Dear editors and reviewers,
Thank you very much for your fruitful and treasured feedback on our paper. Your comments and feedback help enhance the paper's outline and structure. We considered all your comments/reviews in our revised manuscript by highlighting our edits:
Reviewer two
|
Abstract
The abstract is currently not self-explanatory. I suggest revising the abstract to make it more effective and compelling in reporting the unique value of this research and its contribution to theory and practice. First, the goal is unclear. Second, the section on methods is not effectively reported. Finally, the recommendations are not discussed.
|
Thank you for your treasured comment. According to your feedback, we modified and added the contribution and recommendation to the abstract. |
|
Introduction
I believe the introduction leaves out several essential aspects. The authors do not effectively position their research within the current scholarly debate. The authors must clearly position their paper in the scholarly literature, emphasizing how it will contribute to the scientific debate. Improving the article's positioning should lead the authors to identify the knowledge gaps affecting the scholarly debate. At the moment, it is unclear what scientific debate the authors intend to address with their research. The current version of the introduction provides insufficient information to pinpoint the originality and relevance of this research. Furthermore, the authors do not provide a critique of the existing scales and do not explain why a new scale is even necessary. |
Thank you for your treasured comment. Your feedback helped us re-order the introduction in a persuasive method, and we reorganized it to fill the current research gap. Thank you so much. |
|
Literature Review
Authors primarily focus on hypothesis development and various aspects of the scale to be developed in the literature review. However, they make no mention of the scales already in use to measure the variables of interest, their shortcomings, or why a new measurement model is required. Furthermore, the review of literature lacks a global dimension and omits several important studies. Please include the following literature into the study: https://doi.org/10.1016/j.jneb.2014.01.002 https://doi.org/10.3390/ijerph18031175 https://doi.org/10.1016/j.landusepol.2020.105250 https://doi.org 10.1111/cjag.12184 https://doi.org/10.1016/j.jada.2004.08.030 https://doi.org/10.1016/j.jretconser.2021.102556 |
Thank you for your suggestions. We added references that relate to the literature flow. Moreover, regarding that, we did not mention the scale we used or why the new measurement is required: we mentioned the scale adapted from Lee et al. [64] and Ruiz et al. [25] and Petrick [10] in our construct measures part. Moreover, the study added it together in one model to investigate the priority of these measures on the repurchase intentions. Here are the references that we added that match our research scope:
Razzaq, A., Tang, Y., & Qing, P. (2021). Towards Sustainable Diets: Understanding the Cognitive Mechanism of Consumer Acceptance of Biofortified Foods and the Role of Nutrition Information. International Journal of Environmental Research and Public Health, 18(3), 1175. Qing, P., Huang, H., Razzaq, A., Tang, Y., & Tu, M. (2018). Impacts of sellers’ responses to online negative consumer reviews: Evidence from an agricultural product. Canadian Journal of Agricultural Economics/Revue canadienne d'agroeconomie, 66(4), 587-597. Chen, T., Razzaq, A., Qing, P., & Cao, B. (2021). Do you bear to reject them? The effect of anthropomorphism on empathy and consumer preference for unattractive produce. Journal of Retailing and Consumer Services, 61, 102556.
|
|
Materials and Methods The methods section is extremely lacking. Many details are left out, such as how the variables or constructs were developed, how they were contextualized and adapted, and what methods were used to measure relibality and validity. Do the authors employ inter-rater reliability or other techniques? What is the justification for any methods chosen? Furthermore, the details of the data collection process are hazy, and the method is not scientific. I recommend that the authors include as many details as possible to improve the clarity of this section. |
Thank you for your treasured comments. As for the developed variables: The study variables items were taken from prior studies to guarantee the constructs' content validity. We developed our variable by adopting the following constructs from Petrick [10]: behavioural price (4 items), emotional response (5 items), service quality (4 items), and service reputation (5 items). Moreover, Service equity (4 items), confidence benefits (5 items), social benefits (4 items), special treatment benefits (5 items), and repurchase intentions (3 items) are the other constructs adopted from Lee et al. [64] and Ruiz et al. [25]. These variables are developed by considering our level of analysis, "restaurant level," and word consistency remains the main latent variable understandable for customers.
As for study validity and reliability, Table 2 depicts the reliability test (i.e., Cronbach's alpha) for each construct tested between 0.93 and 0.98, above 0.70, indicating the reliability of all variables [74]. All the constructs have substantial internal reliability, as seen by the constructs' composite reliability ranges from .95 to .98 [75]. The structures comprised every item with factor loadings greater than .50 [74]. Regarding discriminant validity, each construct's AVE is higher than the squared correlations between the components (see Table 3) [76].
|
|
Results
The results section is similarly disorganized. The findings are not presented in a way that justifies the study's objectives. Simply including some tables without describing the context or purpose is not helpful to the reader. Please improve the results section and describe in detail how your measurement model evolved and what contribution it makes.
|
Thank you for your comments. According to the results, we ensured that every table embedded in the paper has its own description. Additionally, we ensured that the aims we investigated matched the hypotheses table. For instance, this study aims to fill the current gap by providing new dimensions essential to investigating antecedents of customer repurchase intentions in restaurants (i.e., behavioural price, emotional response, service quality, service reputation, service equity, confidence benefits, social benefits, and special treatment benefits). And we achieved this aim by examining the hypotheses related to the same constructs and the same thing for other purposes. |
|
Discussion and Conclusion
The general discussion merely presents an overview of the study findings, but it does not critically contextualize the study results in light of the extant scholarly debate. As a consequence, the authors are unable to provide us with thick and consistent information about how they are adding to the scholarly debate. Implications for theory and practice are limited. The authors do not provide adequate insights about how we can advance the scholarly knowledge about the new measurement model being proposed. Besides, they do not provide us with compelling management implications supporting decision making processes in the food industry.
Finally, yet importantly, the authors do not effectively elaborate on a relevant and inspiring agenda for further developments.
|
Thank you for your comments. We highlighted our contribution in the main file and the part of the theoretical contribution. Also, we provide the debate according to the highest scholars’ citations discussion in the hospitality and service value field. According to the future research we also highlighted it in the main file, as we proposed new insights regarding our results. Thank you very much for your comments. |
Reviewer 2 Report
The review concerns the article entitled "Service value and repurchase intention in the Egyptian fast-food restaurants: towards a new measurement model”. The subject of the article is interesting and the structure of the article does not raise objections. The review of the literature is extensive and the hypotheses are formulated correctly.
Detailed notes:
1. Please provide in the abstract information about the methods of data analysis.
2. Please include information about the content of individual chapters at the end of the Introduction.
3. When referring to tables in the text, please use numbers instead of words (see, for example, line 356 "Table two ....")
4. Table names should be capitalized (see e.g. line 362 "Table 1. sample…)
5. In Table 3, there are designations for constructs that were not previously explained (e.g. 1, 2, 3, etc.).
6. In the body of the article, references to tables and figures should be written in capital letters (see, for example, line 372).
7. Incorrect p-value values (see e.g. line 375 “…p.000”)
8. The decimal places should be unified throughout the article (sometimes there are 2, other times 3)
Good luck!
Author Response
Responses to Reviewer one
Dear editors and reviewers,
Thank you very much for your fruitful and treasured feedback on our paper. Your comments and feedback help enhance the paper's outline and structure. We considered all your comments/reviews in our revised manuscript by highlighting our edits:
Reviewer one
The review concerns the article entitled "Service value and repurchase intention in the Egyptian fast-food restaurants: towards a new measurement model”. The subject of the article is interesting, and the structure of the article does not raise objections. The review of the literature is extensive, and the hypotheses are formulated correctly.
|
1. Please provide in the abstract information about the methods of data analysis. |
Thank you for your feedback. We added to the abstract the data analysis according to your feedback. |
|
2. Please include information about the content of individual chapters at the end of the Introduction. |
Thank you so much for your comment. We embedded the paper outline in the last introduction according to your feedback. |
|
3. When referring to tables in the text, please use numbers instead of words (see, for example, line 356 "Table two ....") |
Thank you so much for your comment. We changed all the number words to numbers according to your feedback. |
|
4. Table names should be capitalized (see e.g. line 362 "Table 1. sample…) |
Thank you so much for your comment. We capitalized all the table words according to your feedback. |
|
5. In Table 3, there are designations for constructs that were not previously explained (e.g. 1, 2, 3, etc.). |
Thank you so much for your comment. According to your feedback, we added a note below the table referring to the numbers. |
|
6. In the body of the article, references to tables and figures should be written in capital letters (see, for example, line 372). |
Thank you so much for your comment. We modified all the table names according to your feedback. |
|
7. Incorrect p-value values (see e.g. line 375 “…p.000”) |
Thank you so much for your comment. We revised the p-value, and the resultant p value was provided as <0.0001 rather than .000 to be easily interpreted. |
|
8. The decimal places should be unified throughout the article (sometimes there are 2, other times 3) |
Thank you so much for your comment. We unified all the decimals according to your feedback. |
Round 2
Reviewer 1 Report
The authors have made significant revisions in this version and manuscript can be accepted for publication now.